# CAR T Cell Therapy in Hematological Malignancies: Implications of the Tumor Microenvironment and Biomarkers on Efficacy and Toxicity

**DOI:** 10.3390/ijms23136931

**Published:** 2022-06-22

**Authors:** Jing Yuan Tan, Muhammed Haiqal Low, Yunxin Chen, Francesca Lorraine Wei Inng Lim

**Affiliations:** Department of Hematology, Singapore General Hospital Singapore, Outram Road, Singapore 169608, Singapore; jingyuan.tan@mohh.com.sg (J.Y.T.); muhd.haiqal.low@sgh.com.sg (M.H.L.); chen.yunxin@singhealth.com.sg (Y.C.)

**Keywords:** CAR T, tumor microenvironment, biomarkers, hematological malignancies

## Abstract

Chimeric antigen receptor (CAR) T cell therapy has ushered in a new era in cancer treatment. Remarkable outcomes have been demonstrated in patients with previously untreatable relapsed/refractory hematological malignancies. However, optimizing efficacy and reducing the risk of toxicities have posed major challenges, limiting the success of this therapy. The tumor microenvironment (TME) plays an important role in CAR T cell therapy’s effectiveness and the risk of toxicities. Increasing research studies have also identified various biomarkers that can predict its effectiveness and risk of toxicities. In this review, we discuss the various aspects of the TME and biomarkers that have been implicated thus far and discuss the role of creating scoring systems that can aid in further refining clinical applications of CAR T cell therapy and establishing a safe and efficacious personalised medicine for individuals.

## 1. Introduction

Traditionally, cancer treatment is known to be synonymous with chemotherapy, radiation therapy and surgical resections. However, with technological advances and exponential development in scientific research, a growing interest in new forms of cancer treatment such as immunotherapy and, specifically, chimeric antigen receptor (CAR) T cell therapy has emerged [1]. CAR T cell therapy, recognised by the US Food and Drug Administration as a “breakthrough therapy”, has demonstrated remarkable outcomes in certain B cell malignancies [2,3,4,5], and its application is expanding to include myeloma, leukemia and solid organ cancers. Despite the successes that this new cancer therapy has seen, there is still much to understand with regards to its efficacy and safety, and when best to administer this form of therapy to patients.

CAR T cells (CARs) are genetically modified T cells engineered to express receptors that better assist effective targeting of the tumor antigen and subsequent elimination of tumor cells. CARs are typically designed with a single chain fragment consisting of several domains comprising both extracellular and intracellular domains that are important in CAR T cell function [1,2,5,6]. Extracellular domains of CARs are antibody-derived segments directed towards specific tumor antigens. These extracellular domains are able to recognise tumor antigens in a major histocompatibility complex (MHC)-independent manner [7]. This gives CAR T cells an advantage over naturally occurring T cells that recognise tumor cells in an MHC-dependent manner, a mechanism of recognition that is often evaded by tumor cells [7,8]. The intracellular domains on the other hand possess signaling functions that help sustain the immune responses by CAR T cells including activation and proliferation. In the past few years, there has been much focus on improving the design of these CAR T cells, resulting in various intracellular domains being explored to improve the efficacy and safety of these therapies. Currently, there are more than four generations of CARs available. While all generations of CARs make use of CD3ζ as the primary signaling intracellular domain, the first generation CARs only had this as their sole intracellular signaling domain. Even though this generation of CAR T cells were able to trigger immune responses against their intended target, the clinical benefits from these responses were often limited as they neither achieved enough of a level of toxicity to fight the tumors nor were they long lasting owing to their lack of proliferation [9,10]. To overcome these downsides, a co-stimulatory signaling domain (e.g., 41BB, CD28, CD134, etc.) was added into the second generation of CARs and beyond [9,10,11]. This resulted in enhanced cytokine release correlating with increased cytotoxicity, and proliferation was also increased correlating with sustained benefits [10,11]. Overall, the second generation CAR T cells displayed a better immune response. Subsequently in the third generation CARs, more co-stimulatory domains were added in order to improve the function of CAR T cells. While some combined both the 41BB and CD28 signaling domains, others combined 41BB with CD137 [1,2,5,9]. Even though the third generation CARs were developed as an improvement from their predecessor, the clinical advantages they have to offer over second generation CARs are still unclear. Similar to second generation CARs, fourth generation CARs possess a single co-stimulatory domain, but they also contain a protein-encoding domain such as IL-12. The expression of this domain can be induced to further enhance the immune response exhibited by CAR T cells [10].

As of early 2021, over five hundred CAR T cell therapy clinical trials are being conducted worldwide, from just over a hundred at the beginning of 2016. The exponential increase in clinical trials on CAR T cells can be attributed to the success of initial clinical trials [12]. The ultimate goal would be to make this form of treatment personalised to each individual—optimizing efficacy and reducing toxicity. Two main areas that have emerged in recent years to better address this have been the characterization of the tumor microenvironment (TME) and the identification of various predictive and prognostic biomarkers. In this review paper, we discuss the various aspects of the TME and biomarkers that have been implicated thus far, and discuss the role of creating scoring systems that can aid in further refining clinical applications of CAR T cell therapy and establishing a safe and efficacious personalised medicine for individuals.

## 2. Biomarkers and Tumor Microenvironment

Biomarkers represent an important aspect of precision medicine. They allow the objective characterization of biological processes. This method of study has proven to be more advantageous given that medical symptoms can vary significantly from patient to patient and are potentially absent in certain cases. Biomarkers are objective, accurate and reproducible within an entire population [13]. According to the National Institute of Health’s Biomarkers Definitions Working Group, a biomarker is defined as “a characteristic that is objectively measured and evaluated as an indicator of normal biological processes, pathogenic processes, or pharmacologic responses to a therapeutic intervention” [14]. By this definition, a biomarker is not limited to the mere understanding of a disease, but can serve as a predictor, or as a prognostic and treatment marker. There are various different categories of biomarkers, each providing different information about a disease process. These include diagnostic, prognostic, predictive and response biomarkers and more. Table 1 summarizes the different types of biomarkers as well as each ones scientific value [15]. In CAR T cell therapy, where disease and response are heterogenous, there is an increasing need and importance to find novel biomarkers within the product and host that can better assess immune characteristics pre and post treatment with the ultimate aim of optimizing efficacy and reducing the severity of toxicity.

Aside from biomarkers, analyzing the host characteristics and tumor microenvironment are also important. These characteristics vary greatly between different individuals and different tumor types [16]; thus, understanding the interactions of the TME with the CAR T cell product will aid in achieving personalised therapy. The tumor microenvironment (TME) consists of a diverse set of components that collectively contribute towards the interplay of pro-immune and immunosuppressive signaling [17]. Assessment of the TME before and after CAR T therapy may be beneficial in predicting the response, persistence and toxicities.

Routine laboratory tests are the first line when it comes to the identification of biomarkers that can predict efficacy and toxicity. However with an increasing number of novel biomarkers and the need to characterize a patient’s TME, more specialized laboratory work and tests are needed, including the use of flow cytometric analysis, polymerase chain reaction (PCR) for transcriptomic studies and enzyme-linked immunosorbent assay (ELISA) for proteomics study. As we continue to further research in these fields, it is also important to ensure that these findings can be universally applied, especially in less medically advanced countries. The next sections discuss the role of the TME and biomarkers in the CAR T cell’s response, resistance and toxicity.

## 3. Role of Tumor Microenvironment and Biomarkers in CAR T Cell Response and Resistance

The response to CAR T cell therapy varies between hematological malignancies. The best response has been seen in adult B cell acute lymphoblastic leukemia (B-ALL) patients with complete remission (CR) rates ranging from 83% to 93% [18,19,20,21]. Less optimal responses were seen in patients with diffuse large B cell lymphoma (DLBCL), with CR rates between 39% and 54% [22,23,24,25]. CR rates were even lower in chronic lymphocytic leukemia (CLL) patients, ranging between 20% and 30% [26,27,28]. Aside from initial response rates, the duration of response is equally important and has been shown to, likewise, vary between hematological malignancies treated. Interestingly, while B-ALL patients had higher initial response rates, their duration of response seemed lesser than patients with DLBCL and CLL [29,30]. Numerous emerging studies detailing outcomes of patients with hematological malignancies have observed that while initial response rates were good, the risk of relapse was high. An increasing amount of work has focused on dissecting the tumor microenvironment and discovering biomarkers that can predict the response, persistence and resistance to CAR T cell therapy. We review the current available evidence in this section.

### 3.1. Tumor Microenvironment

Postulated mechanisms for resistance were reviewed by Lemoine et al. with a focus on three aspects—CAR T cells (e.g., lack of expansion and defective effector function (exhaustion), the tumor microenvironment and the cancer cells (e.g., loss of target antigen and expression of inhibitory ligands (PD-L1 expression) [31]. While the relevance of TME may be more obvious in solid organ cancers due to hypoxia/metabolism-related factors and tumor trafficking, it is the heterogenous population of immunosuppressive cells and acellular elements such as immunosuppressive cytokines that affect the response of hematological malignancies to CAR T cell therapy. The diverse set of components in the TME interact with each other and create a balance between pro-immune and immunosuppressive signaling. CAR T cells proliferate and expand in the recipient patient in response to in vivo signals. As such, the functions of CAR T cells are susceptible to the immunosuppressive nature of the TME. Consequently, many groups have explored the individual roles of the different cellular and acellular elements implicated in CAR T cell inhibition that affect its therapeutic efficacy. We look at three important cells in this section—myeloid-derived suppressor cells, tumor-associated macrophages and regulatory T cells.

### 3.2. Immunosuppressive Cells

Myeloid-derived suppressor cells (MDSCs) are immature myeloid cells that arise from bone marrow myeloid progenitors [32]. These immature myeloid cells differentiate into mature cells in healthy adults, but in pathological conditions where levels of inflammation are high, the differentiation process is interrupted with the consequent expansion of a heterogenous clump of immature myeloid cells including immature macrophages, immature granulocytes and immature dendritic cells [33,34]. These cells are able to suppress both the innate and adaptive response: specifically, the suppression of T cell function [35]. MDSCs are also major sources of reactive oxygen and nitrogen species, which are harmful to T cells [36]. In cancer patients, tumor cells release signals that recruit these MDSCs and subsequently expand them, resulting in a tumor-promoting milieu. The inhibitory effect of MDSCs on CAR T cells has been mainly demonstrated in patients with solid organ tumors, including breast, liver and prostate cancer and sarcoma [37,38,39,40]. Studies of MDSCs in hematological malignancies undergoing CAR T cell therapy have been considerably fewer. One study by Jain et al. observed that high blood levels of monocytic MDSCs in patients with B cell lymphomas prior to CAR T cell therapy resulted in a lack of durable response [41]. Aside from its direct inhibitory effect on T cells, the interaction of MDSCs with immune checkpoint molecules such as PD-L1 can induce T cell anergy and apoptosis as was demonstrated by Groth et al. [42]. MDSCs also promote the maturation of tumor-associated macrophages (TAMs) and regulatory T cells (Tregs) that further contribute to the immunosuppressive TME. Tumor-associated macrophages (TAMs) comprise a heterogenous population of phenotypically and functionally distinct macrophages that have both antitumor roles and tumor-supporting roles [43,44,45]. They enable tumor growth through the secretion of growth factors, proangiogenic factors and matrix degradation enzymes [46,47,48]. At the same time, they inhibit tumor growth through the secretion of antiinflammatory cytokines such as IL-10 or through the expression of inhibitory immune checkpoint ligands such as PD-L1 that can suppress immune cells. A higher TAM density in hematological malignancies has been correlated with poorer survival outcomes [49,50]. In the context of CAR T cell efficacy, Yan et al. demonstrated that in patients with refractory B cell Hodgkin lymphoma receiving anti-CD19 CAR T cell therapy, increased TAM infiltration was negatively associated with remission status [51]. Finally, we look at regulatory T cells (Tregs). These cells co-express CD4, CD25, transcription factor Forkhead box protein 3 (FOXP3) and CD127 and make up a small portion of the CD4+ T cell population [52]. They maintain immune homeostasis and play an important role in immune tolerance by inhibiting activated effector T cells through the secretion of cytokines such as IL-10 and IL-35, as well as through contact-dependent inhibition such as binding to CTLA-4 [53,54]. In solid organ tumors, increased circulating or tumor-infiltrating Tregs have been associated with poorer patient survival [55,56]. On the other hand, published data on the prognostic significance of Tregs in hematological malignancies are conflicting [57]. While there have been no specific studies evaluating the inhibitory role of Tregs in patients with hematological malignancies receiving CAR T cell therapy, Duell et al. demonstrated that in patients with relapsed/refractory B-ALL receiving blinatumomab (a bispecific CD3/CD19 T cell engager), responders had significantly fewer Tregs than non-responders. Additionally, the depletion of Tregs in non-responders restored T cell proliferation [58]. In another study involving adoptively-transferred cytotoxic T lymphocytes (CTLs) in in vivo models of acute myeloid leukemia, Zhou et al. showed that depleting Tregs prior to transfer resulted in the enhanced proliferation of CTLs and increased their therapeutic benefits [59]. Overall, the above studies illustrate that characterizing the TME of patients prior to and after receiving CAR T cell therapy can predict the response, where high levels of MDSCs, Tregs and TAMs would likely be met with reduced efficacy and persistence.

Moving forward, functional assessment of the interplay between the tumor, TME and CARs will also allow further understanding of the factors promoting or inhibiting T cell trafficking and infiltration into tumor sites. In cellular therapeutics, this interaction needs to be assessed at multiple levels—at the tissue architectural level as well as at the single-cell level. Spatial profiling of the immune cells with concurrent single-cell level proteomic and transcriptomic profiling has started to provide a peek into this dynamic interaction [60]. Imaging modalities can also assist in providing spatial assessment of tumor–CAR T interaction. Novel non-immunogenic reporters in PET imaging can now be deployed to trace CAR T cells to provide a real-time assessment of the in vivo distribution and fate of CAR T cells [61].

### 3.3. Biomarkers

Biomarkers can also aid in predicting the response to CAR T cell therapy. When thinking about biomarkers, patient characteristics and disease markers in the form of laboratory tests are first to come to mind as they are readily available and easy to perform. One such biomarker is lactate dehydrogenase (LDH). A marker correlated with high tumor burden, studies have consistently demonstrated an association between its higher levels and poorer outcomes in patients with B cell malignancies receiving CAR T cell therapy [62,63,64]. Garcia et al. also demonstrated the possibility of employing risk indexes to predict outcomes. In a study involving R/R DLBCL patients treated with CAR T cell therapy, a higher age-adjusted international prognostic score (aaIPI) was associated with poorer progression free survival (PFS) and overall survival (OS). High-risk IPI was associated with poorer PFS [65].

However, these biomarkers and risk scores may not be universally applicable to all hematological malignancies, and, thus, the exploration of other biomarkers that may influence the efficacy of CAR T cells is crucial. In this regard, the identification of molecules that play a role in proliferation, differentiation potential, effector function and exhaustion hold promise for optimizing the proliferative capacity and antitumor activity. We detail some of these below.

The percentage composition of T cell subsets can influence CAR T cell function. T cell subsets are grouped according to differentiation levels and can be distinguished based on the presence of different surface markers (e.g., CD45RA, CD45RO, CD27, CD28). These subsets include stem cell memory (T_scm_), central memory (T_cm_), effector memory (T_em_), effector memory that re-express CD45RA (T_emra_) and effector (T_eff_) T cells. Better CAR T cell therapy outcomes have been observed in patients with a higher proportion of less differentiated T cell subsets in the CAR T cell product [66,67,68]. This is likely due to these less differentiated T cells having higher expansion capabilities and potential to differentiate into other T cell subsets such as T_cm_ and T_eff_ that have both persistence and cytotoxic capabilities. Garfall et al. observed that leukapheresis products from multiple myeloma patients that had higher frequencies of CD8+CD45RO-CD27+ memory subset, had better proliferative capabilities and better responses [69]. In an experimental study, Sabatino S. et al. genetically engineered and generated CD19-CAR-modified CD8+ T_scm_ and demonstrated that when compared to CD8+ CAR T cells generated by current clinical protocols, CD19-CAR-modified CD8+ T_scm_ exhibited enhanced metabolic fitness and longer lasting antitumor effects against ALL xenografts [70]. Thus, creating a CAR T cell product with a greater proportion of memory T cells and a lower proportion of effector T cells can help to optimize efficacy and persistence.

Immune checkpoint inhibitory molecules also serve as important biomarkers to predict CAR T cell therapy efficacy and persistence. The most widely studied have been PD-1, LAG-3 and TIM-3, which correlate with T cell exhaustion and, consequently, a poor response to CAR T cell therapy [71,72,73]. Mechanisms include the inhibition of T cell expansion and cytokine release and, consequently, reduced cytotoxicity. In patients with chronic lymphocytic leukemia, Fraietta et al. demonstrated that patients who achieved complete remission had lower percentages of PD1+CD8+ CAR T cells pre-infusion than those who achieved only partial remission/non-responders [66]. In children with B-ALL, Li et al. demonstrated that children who were treated with a combination CAR T cell therapy and PD-1 inhibitors (pembrolizumab or nivolumab) had better outcomes and improved persistence of CAR T cells [74]. Similar findings were observed in solid tumors [75]. In patients with large B cell lymphomas, Deng et al. observed that patients who progressed after axi-cel still had detectable CAR+ T cells, but more than half of these CD8 CAR T cells were LAG3+ and TIM3+. Seemingly, patients who did not achieve a robust early molecular response had a higher frequency of CD8 T cell exhaustion signatures that were characterized by an increased proportion of LAG3+TIM3+ cells [76]. Supporting these observations are several studies that have shown that the blockade of PD-1, TIM-3 and LAG-3 can boosts the effector functions of CAR T cells [77,78]. Recent evidence has also, however, suggested that inhibitory receptors alone are not sufficient to distinguish between exhausted and activated T cells as these receptors are similarly upregulated in T cell activation as a mechanism of modulating co-stimulatory signaling [79]. Exhausted T cells have significant epigenetic organization and distinct transcriptional signatures that translate into identifiable cell surface makers and constitute a potential area for the identification of exhaustion and for predicting efficacy and persistence. Some of these markers include CX3CR1, CD38 and CD101; this was reviewed by Gumber [80].

Cytokines are another group of novel biomarkers that have gained attention and spurred modifications to CAR T cell products that have enhanced proliferation, and are able to revert T cell exhaustion and promote antitumor abilities. Multiple inflammatory cytokines such as IL-6, IL-7, IL-8, IL-12, IL-15, IL-18, IFN-y and TNF-a have been shown to be able to enhance the cytotoxic functions of T cells and NK cells [64,81,82,83,84]. Harnessing this knowledge, groups have modified CAR T cells to secrete cytokines such as IL-12 and IL-18 and observed that there was better tumor eradication and CAR T cell persistence [85,86]. On the other hand, IL-10, TGF-B and IL-4 are immunosuppressive cytokines that can contribute to CAR T cell dysfunction [87,88]. These cytokines can either directly inhibit the effector function of CAR T cells or can recruit and activate MDSCs and Tregs that can affect CAR T cell function as previously mentioned.

Combinations of patient and disease characteristics, laboratory tests and knowledge of more specific molecules can serve as important biomarkers in predicting therapy efficacy as well as for the future development of “armored” CAR T cells that have better efficacy.

## 4. Tumor Microenvironment and Biomarkers in CAR T Cell Toxicity

Despite the remarkable success of CAR T cell therapy, the incidence of CAR T-associated toxicities are high and represent a significant limitation to this form of therapy. These toxicities can be severe and fatal. Cytokine release syndrome (CRS) and immune effector cell-associated neurotoxicity syndrome (ICANS) are two important toxicities in CAR T cell therapy. The pathophysiology of both conditions has been reviewed by Siegler and Kenderian [89].

Briefly speaking, CRS occurs due to activated CAR T cells triggering an inflammatory response of varying degrees. Symptoms include fever, headache, myalgia, malaise and, in severe cases, multiorgan dysfunction, hypotension requiring inotropic support and hypoxia requiring mechanical ventilation. Inflammatory cytokines such as tumor necrosis factor (TNF)α and interferon (IFN)γ are released and in turn activate monocytes and macrophages to release more cytokines including IL-1 and IL-6. Correspondingly, IL-6 levels are highly elevated in patients with CRS and treatment includes IL-6 inhibitors such as tocilizumab [90]. CRS severity is graded according to the ASTCT grading scale and its incidence has been reported as close to 100% of varying severity in CART 19 clinical trials [91,92,93,94]. ICANS also occurs as a result of activated CAR T cells triggering an inflammatory response. However, in addition, the systemic inflammation activates endothelial cells, which drive blood–brain barrier (BBB) dysfunction. BBB dysfunction results in increased permeability, allowing cytokines to accumulate in the cerebrospinal fluid causing neurotoxicity [95]. Symptoms reported include confusion, delirium, encephalopathy and cognitive dysfunction often associated with language dysfunction, which manifests as word finding difficulties, handwriting disturbances or, in severe cases, mutism. In severe cases, patients can lose consciousness requiring mechanical ventilation and the most feared neurological complication is cerebral edema, which is invariably fatal. Its incidence is lower than in CRS, varying anywhere between 5% and 70% [96].

Because of the high incidence and potential severe morbidity of both CRS and ICANS, predictive biomarkers for these toxicities are important. Identifying such biomarkers can allow early recognition, appropriate counselling to patients and early treatment. In addition, understanding the TME and its impact on these toxicities can pave the way for future strategies to optimize the TME and hopefully reduce the risk of CAR T toxicities.

### 4.1. Tumor Microenvironment

As aforementioned, the TME contains a heterogenous population of cells that can provide both immunosuppressive and immunomodulatory signals. In the context of CAR T cell toxicity, pro-immune signaling can trigger the proliferation of CAR T cells and the consequent secretion of cytokines that create a proinflammatory TME and unintentionally potentiate adverse events. Macrophages and endothelial cells have been studied with regards to the role they play in the pathogenesis of CAR T toxicities.

Macrophages, when activated, secrete IL-1, IL-6, nitric oxide (NO) and catecholamines that result in inflammation and contribute to CRS/ICANS. In humanized mice with high leukemic burden, Norelli et al. observed that it was the human monocytes rather than CAR T cells that were the major source of IL-1 and IL-6. Monocyte depletion had a positive effect on preventing CRS but negatively impacted in vivo CAR T cell expansion [97]. The activation of macrophages by CAR T cells is through the secretion of granulocyte-macrophage colony-stimulating factor (GM-CSF). In the ZUMA-1 trial, severe neurotoxicity was found to be associated with high levels of GM-CSF [95]. Subsequent studies by Sachdeva et al. and Sterner et al. demonstrated that the inactivation/neutralization of GM-CSF in CAR T cells reduced toxicities through abolishing the monocyte-dependent release of CRS mediators [98] and reducing myeloid and T cell infiltration in the central nervous system, consequently reducing neuroinflammation [99]. In both studies, the reduction in GM-CSF did not affect CAR T function but, on the contrary, enhanced its antitumor activity. The activation of endothelial cells in the TME by the release of inflammatory cytokines from engaged CAR T cells has also been implicated in the pathophysiology of CRS and ICANS. Activated endothelial cells secrete cytokines such as IL-6 and IL-8 and proinflammatory factors such as monocyte chemoattractant protein-1 (MCP-1), coagulation cascade activator plasminogen activator inhibitor-1 (PAI-1) and vascular endothelial growth factor (VEGF) [100,101,102] that further amplify inflammatory responses. These activated endothelial cells also release angiopoietin-2 (Ang-2) and von Willebrand factor (vWF), which causes endothelial permeability [103,104]. Direct injury by cytokines further contributes to vascular permeability causing leakage, organ hypoperfusion and, consequently, dysfunction [95]. Because the endothelium is the most important component of the blood–brain barrier (BBB), endothelial dysfunction and permeability allow an extensive number of immune cells and cytokines to infiltrate the CNS [105,106]. These exacerbate the inflammatory cascade in the CNS that had already been activated by resident proinflammatory cells. Consequently, patients develop cerebral edema, microthrombi and disseminated intravascular coagulopathy, signs that have been observed in ICANS. Autopsies of patients with ICANS revealed intravascular VWF binding and platelet microthrombi [107].

Understanding the role of these cells and proinflammatory factors in the TME in predicting the severity of CRS and ICANS will be integral in future strategies targeted at reducing the density of these proinflammatory factors to reduce toxicity. Designing treatment interventions targeted at these factors may subsequently add to our armamentarium of available treatment for CAR T cell toxicities.

### 4.2. Biomarkers

LDH, c-reactive protein (CRP) and ferritin have been shown to be associated with the development of CRS and ICANS [108,109]. In a study of 15 patients, higher levels of CRP, ferritin and D-dimer were demonstrated by Hu et al. to be correlated with more severe CRS, and lower levels predicted a better response to tocilizumab and corticosteroids [110]. A larger study with 51 patients, while also showing an association of CRP and ferritin with CRS, failed, however, to predict the development of severe CRS [111]. Thus, more specific biomarkers are needed with higher sensitivity and specificity. One option is to combine biomarkers into a scoring system. Pennisi et al. proposed a modified EASIX (endothelial activation and stress index) score that calculated peri-CAR T cell infusion, this could discriminate patients who developed severe (grade ≥ 3) CRS and ICANS [112]. The EASIX score (lactate dehydrogenase (LDH; U/L) X creatinine (mg/dL)/platelets (PLTs; 109 cells/L)) is a marker of endothelial damage that correlates with outcomes in allogeneic hematopoietic cell transplantation [113]. In the modified EASIX score, creatinine is replaced with CRP. Another group, Greenbaum et al., added ferritin and CRP to the EASIX score and found that EASIX-F was correlated with CRS (grade 2–4) and EASIX-FC was correlated with ICANS (grade 2–4) [114]. The manifestation of biochemical abnormalities on laboratory tests may, however, lag behind the emergence of inflammatory cytokines in the blood. Thus, identifying and understanding the various inflammatory cytokines that harbor the onset of severe CRS and ICANS is crucial and can allow even earlier detection and treatment.

In CRS, IL-6, IL-10 and IFN-y have been shown to be the strongest contributors to its development. Other cytokines implicated include IL-2, IL-8, IL-12, TNF-a, monocyte chemotactic protein (MCP-1) and macrophage inflammatory protein 1a, which have also demonstrated predictive value [97,115,116]. Higher concentrations of these cytokines were observed in patients with a higher grade of CRS. These inflammatory signals further interact amongst themselves, which triggers a constant loop of cytokine generation. For example, IFN-y secreted by the activation of T cells and tumor cells can stimulate other immune cells such as macrophages to secrete more proinflammatory cytokines (e.g., IL-1, IL-6, IL-8, IL-12, IL-15, IL-1RA, TNF-a) [107,117,118]. These inflammatory markers have also been implicated in ICANS. Some studies have demonstrated that concentrations of IL-8, Il-10 and MCP-1 in cerebrospinal fluid correlate with the severity of ICANS [107,119,120].

## 5. Concluding Remarks and Future Directions

With increasing efforts to “perfect” CAR T cell therapy, a deeper understanding of the tumor microenvironment and identifying predictive biomarkers will be paramount. Given the complexity of the immune system, it is not a single component in the TME or a single biomarker that will be able to adequately predict efficacy and toxicity but rather a combination. Understanding the interplay between the biomarkers, TME, patient and disease factors and the CAR T cell product will be integral in paving the way to devise solutions to overcome current limitations. Engineering CARs that are resistant to immunosuppressive factors in the TME or CARs that are able to provide immunostimulatory signals in the form of stimulatory cytokines that increase survival, proliferation and the antitumor activity of T cells, and rebalance the tumor microenvironment will help to increase the efficacy of T cells. Harnessing the role of predictive biomarkers will also aid in predicting efficacy and, importantly, aid in the early identification of toxicities to allow closer monitoring and early treatment. Understanding the TME and identifying biomarkers that can assess the response and toxicity will greatly aid in possibly identifying patients who might benefit most from these therapies in the near future.

## Figures and Tables

**Table 1 ijms-23-06931-t001:** The different types of biomarkers as well as each ones scientific value [15].

Type of Biomarker	Scientific Value	In the Context of CAR T Cell Therapy
**Diagnostic**	To confirm the presence of a disease and the extent of a specific subset	Assess CAR T cell product characteristics (e.g., T cell quality, proportion of phenotypes) pre-infusion
**Prognostic**	To identify the likelihood of clinical outcomes such as disease progression or overall survival	Assess reasons for CAR T cell therapy resistance and disease relapse (e.g., loss of target antigen, expression of inhibitory ligands)
**Predictive**	To identify individuals who are more likely to benefit from a certain type of therapy	Assess patient and disease characteristics pre and post CAR T cell therapy to predict response and risk of toxicity
**Response**	To show that a biological response has occurred from exposure to treatment	Assess function of CAR T cell therapy through identification of biomarkers that can measure host immune response to cell therapy
**Safety**	To indicate the presence or extent of toxicity related to treatment	Aid in early identification and quantification of severity of CAR T cell-related toxicities (e.g., measuring cytokines)

## Data Availability

Not applicable.

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
