# Peer review of "CAR T Cell Therapy in Hematological Malignancies: Implications of the Tumor Microenvironment and Biomarkers on Efficacy and Toxicity"

_ijms, 2022, doi:10.3390/ijms23136931_

Round 1

Reviewer 1 Report

Nice paper. I would recommend it's publication, pending English proofing.

Author Response

We thank the reviewer for the comment.

Reviewer 2 Report

In this review, the authors discuss the various aspects of the tumor microenvironment and biomarkers that have been implicated so far and the role of creating scoring systems that can help further refine clinical applications of CAR T cell therapy and establish safe personalized medicine and effective for individuals.

The introduction is clear as is the structure of the entire article, but such articles already exist in the literature.

The article does not contain the totality of current knowledge, the references are few and not very recent, in fact it does not even contain information that is current, important and useful for the topic dealt with such as those collected in this recent article: "Davila ML, Brentjens RJ. CD19-Targeted CAR T cells as novel cancer immunotherapy for relapsed or refractory B-cell acute lymphoblastic leukemia. Clin Adv Hematol Oncol. 2016 Oct; 14 (10): 802-808. PMID: 27930631; PMCID: PMC5536094 . "

Therefore I believe that this article cannot be considered for publication on IJMS unless completely revised.

Author Response

We thank the reviewer for the comment. We agree that articles on tumor microenvironment (TME) and biomarkers for CAR T cell therapy do exist in the literature. These articles are broad with regards to CAR T cell therapy encompassing solid tumors and haematological malignancies in the same review. We believe our article that focuses on haematological malignancies would allow readers to have a specific article to reference to when thinking about how TME and biomarkers influence CAR T cell therapy in haematological malignancies. Tumor microenvironment in CAR T cell therapy is more described in solid-organ malignancies including other factors like trafficking and hypoxia. In our review paper we focused the TME on haematological malignancies and summarised the current available evidence on immunosuppressive cells in the TME on CAR T therapy. 

In addition, to our knowledge, other review papers have not mentioned about potential scoring systems that have arisen in the past 1-2 years which have been mentioned in our review paper, i.e modified aaIPI score, EASIX score, EASIX-FC score. Through our review article, we spur further research on development of additional scoring system that can aid in predicting not just toxicity but for clinical efficacy as well. Additionally, we also mentioned the role of imaging in functional assessment (line 223-232) which is not mentioned in other review papers. We also believe that our references are current with most references being published in the past 5 years. However, we have also repeated a search on TME and biomarkers with CAR T cell therapy in haematological malignancies and added additional references including studies from the reference that the reviewer has suggested mainly highlight CRP and tumor burden as a predictor for toxicity. CRP as a predicting for toxicity has been mentioned adequately in our review paper (Line 383 to 387)

Thus, we believe that our article as it stands is still a meaningful article to be published.

Reviewer 3 Report

The manuscript by Tan et al. describes the importance of the tumor microenvironment (TME) and biomarkers on CAR T cell therapy. Various aspects of the TME and biomarkers are discussed. This is important for improving CAR T cell responses with less severity of toxicity.

The manuscript is well-structured, some typos should be corrected.

The reference in line 128 and 174 is missing.

In line 157 and 261 first name should be replaced by surname.

Author Response

We thank the reviewer for the comment. Reference to line 128 and 174 have been added. The names for line 157 and 261 have also been changed

Round 2

Reviewer 2 Report

Now it's ok for journaling